# Short-Term Thermal Stress Affects Immune Cell Features in the Sea Urchin *Paracentrotus lividus*

**DOI:** 10.3390/ani13121954

**Published:** 2023-06-11

**Authors:** Carola Murano, Alessandra Gallo, Aurora Nocerino, Alberto Macina, Stefano Cecchini Gualandi, Raffaele Boni

**Affiliations:** 1Department of Integrative Marine Ecology, Stazione Zoologica Anton Dohrn, Villa Comunale, 80121 Naples, Italy; carola.murano@szn.it; 2Department of Biology and Evolution of Marine Organisms, Stazione Zoologica Anton Dohrn, Villa Comunale, 80121 Naples, Italy; alessandra.gallo@szn.it (A.G.); auroranocerino@outlook.it (A.N.); 3Unit Marine Resources for Research, Stazione Zoologica Anton Dohrn, Villa Comunale, 80121 Naples, Italy; alberto.macina@szn.it; 4Department of Sciences, University of Basilicata, Via dell’Ateneo Lucano, 10, 85100 Potenza, Italy; stefano.cecchini@unibas.it

**Keywords:** coelomic fluid, osmolarity, pH, coelomocytes, reactive oxygen species, respiratory burst, mitochondrial membrane potential, lipid peroxidation

## Abstract

**Simple Summary:**

The immune response to pathogens is one of the organic functions at risk due to climatic change. The effects of a short-term increase in temperature on immune cell functions have been evaluated using a simple animal model, such as the sea urchin, exposed to temperatures above its comfort zone (17 °C), causing moderate and severe heat stress. After 3 and 7 days of exposure to the three temperatures tested, the coelomic fluid of the animals was collected. The coelomocytes were isolated and typed and the mitochondrial activity, the amount of lipid peroxidation, and the production of hydrogen peroxide were evaluated. These cells were also stimulated with phorbol 12-myristate 13-acetate (PMA) to evoke respiratory burst, a marker of immune response. Heat stress altered both the distribution of the various types of coelomocytes and their mitochondrial activity; it also increased lipid peroxidation and reduced the production of hydrogen peroxide. The respiratory burst occurred in the coelomocytes of all three experimental groups; however, it was greater in the group kept at 17 °C. In conclusion, thermal increase alters immune cell functions with possible repercussions on defense mechanisms against pathogens.

**Abstract:**

Due to global warming, animals are experiencing heat stress (HS), affecting many organic functions and species’ survival. In this line, some characteristics of immune cells in sea urchins subjected to short-term HS were evaluated. *Paracentrotus lividus* adult females were randomly divided into three groups and housed in tanks at 17 °C. In two of these tanks, the temperatures were gradually increased up to 23 and 28 °C. Celomatic fluid was collected after 3 and 7 days. The coelomocytes were morphologically typed and evaluated for their mitochondrial membrane potential (MMP), lipoperoxidation extent (LPO), and hydrogen peroxide content (H_2_O_2_). Respiratory burst was induced by treatment with phorbol 12-myristate 13-acetate (PMA). HS caused a significant change in the coelomocytes’ type distribution. MMP increased in the 23 °C-group and decreased in the 28 °C-group at both 3 and 7 days. LPO only increased in the 28 °C-group at 7 days. H_2_O_2_ progressively decreased together with the temperature increase. Respiratory burst was detected in all groups, but it was higher in the 17 °C group. In conclusion, the increase in temperature above the comfort zone for this animal species affects their immune cells with possible impairment of their functions.

## 1. Introduction

Global warming is a serious threat to the survival of many plant and animal species on our planet [1]. In animals, besides reproduction [2], thermal increases have been demonstrated to have harmful effects on the respiratory and cardio-circulatory systems [3], as well as on systems capable of maintaining the homeostasis of the body’s functions [4] and the immune system [5,6]. An organism’s response to heat stress (HS) is influenced by its magnitude and duration and by other environmental parameters (e.g., humidity, ventilation, etc.) associated with stress as well as strategies to escape heat stress or thermotolerance of the species and the individual [7]. A further aspect to consider is the species-related capacity to regulate body temperature and, in aquatic animals, many additional temperature-related effects such as the solubility of oxygen and pollutants, the toxicity of chemicals, and the seawater pH, density, and electrical conductivity should be considered (for a review see [8]). Moreover, the thermal increase may occur as a stable and long-lasting increase in the average temperature or as heat waves [9]. The effect of the latter could be further unfavorable by more deeply undermining the contrast capabilities produced as a reaction of the individual.

In marine invertebrates, the effects of HS have been mostly evaluated in terms of the survival of the entire organism or single cells as well as in relation to reproductive activity. In a previous study [10], we evaluated the impact of long-term (30 days) HS on the sperm quality of *Mytilus galloprovincialis*, disclosing morphological and functional sperm alterations associated with HS. In the sea urchin *Arbacia punctulata*, short-term (7 days) HS decreased gamete production and increased reactive oxygen and nitrogen species (ROS and RNS, respectively) content in the gonadal tissue [11]. 

The effects of thermal stress on immune activity are an interesting object of study considering that the survival of marine animals under global warming strongly depends on the sensitivity of these organisms to pathogens [12,13]. Interestingly, the immune cells from sea urchins have been indicated as sentinels of environmental stress as well as valid tools to uncover basic molecular and regulatory mechanisms of immune response, supporting their use in immunological research [14]. 

Coelomocytes are free-floating cells, belonging to different cell types, present in the coelomic liquid but also found among the tissues of several body parts of echinoderms and assigned to perform various activities including nutrient transport and immune defense [15]. The immune defense activity involves phagocytosis but also the release of bioactive molecules with antimicrobial activity. Among these, the production of hydrogen peroxide (H_2_O_2_) undoubtedly plays a primary role and makes these cells similar to the neutrophil granulocytes and macrophages of mammals [16,17]. H_2_O_2_ production in coelomocytes involves both the quiescent and stimulated states but increases in the latter [16]. Following immune cell activation, the nicotinamide adenine dinucleotide phosphate (NADPH) oxidase catalyzes the reduction of O_2_ to O_2_^−^, which is further transformed by dismutation to H_2_O_2_ [18]. This feature may represent a useful biomarker to evaluate the immune competence of echinoderms under stress conditions. The effects on the immune response following 14 days of exposure to 20 (control), 25 and 30 °C were evaluated in the subtidal *Lytechinus variegatus* and the intertidal *Echinometra lucunter* sea urchins [19]. Although both species exhibited significant temperature-associated changes in the number of red and white amoebocytes, only *L. variegatus* showed a significant decrease in phagocytic activity, cell adhesion, and cell spreading. In the Pacific sea urchin *Strongylocentrotus purpuratus*, the detection of two pathologies that were extremely rare or absent in northern California but common at warmer sites in southern California were associated with an increase in seawater temperature [12]. Further studies in microcosm demonstrated that increased seawater temperature in diseased individuals was associated with significantly larger lesions and a significantly lower gonadal index.

In this study, the effects of short-term HS in a homeothermic organism, such as the sea urchin, were evaluated through the assessment of some physiological characteristics of the coelomocytes. In particular, mitochondrial activity, lipid peroxidation, and (H_2_O_2_) production were monitored in these cells after exposure to short-term HS using fluorescence spectrometry techniques. The observed effects were further validated by directly evoking respiratory burst, a key defense mechanism against a wide range of pathogens occurring in immune cells and consisting of the rapid release of reactive oxygen species [20]. 

## 2. Materials and Methods

### 2.1. Materials

Unless otherwise indicated, all materials used for this study were purchased from Merck Life Science (Milan, Italy) and cell culture tested.

### 2.2. Animal Collection and Breeding Care

Adult individuals of *P. lividus* were collected by the personnel of the Infrastructure for Marine Research Unit of the Stazione Zoologica Anton Dohrn in the Gulf of Naples in the middle of January 2023 from an area neither privately owned nor protected, according to the authorization of Marina Mercantile (DPR 1639/68, 19 September 1980, confirmed on 10 January 2000) and transported inside thermal boxes.

Although no authorization is required for experimental studies on sea urchins, all animal procedures were applied in compliance with the guidelines of the European Union (directive 2010/63/EU and following D. Lgs. 4/03/2014 n.26) on the protection of animals used for scientific purposes by reducing the number of specimens used and any pain or stress on animals. 

After collection, the sea urchins were transported to the Marine Biological Resources Service, where they were acclimated for 7 days before use in tanks (1 animal/5L) with running natural seawater under the following conditions: a temperature of 18 ± 2 °C, pH 8.1 ± 0.1, a salinity of 39 ± 0.5 ppm, and a photoperiod of 10 h L: 14 h D. During this period, sea urchins were fed with fresh green algae *Ulva* sp. After acclimatization, the female sea urchins were screened based on morphological analysis [21] and used for the experimentation. The decision to limit the analysis to female individuals only was linked to experimental reasons, such as avoiding interactions of a reproductive nature, which could interfere with the results, and reducing the variability of responses within the experimental groups, considering sex-based differences in the expression of genes related to the immune and stress responses in coelomocytes, as detected in the sea urchin *Arbacia lixula* [22].

### 2.3. Experimental Design

A total number of 36 selected female sea urchins were enrolled in this study. The first group of 18 animals was randomly divided into three open-circuit seawater tanks assuring up to three complete water changes daily. In the first tank, seawater was maintained at an ambient seawater temperature (17 °C), whereas, in the other two tanks, the seawater temperature was gradually increased to 23 °C (moderate HS) and 28 °C (severe HS) over 5 days. Then, the experiment started and lasted 7 days. During the experimental period, the animals were not fed to avoid contamination by feces and food residues that would have involved cleaning operations, thus compromising the rearing environment and causing a further stress source. The first sampling was carried out three days after the experiment started. Three animals from each experimental group underwent coelomic fluid sampling; after which, these animals were not used for further experimentation. After 4 days, coelomic fluid was collected from the remaining three animals from each group. The experiment was replicated one more time using the same number of animals.

### 2.4. Coelomic Fluid and Cell Sampling

Coelomic fluid was withdrawn by a 26-gauge needle attached to a sterile syringe that was inserted through the peristomium into the coelom cavity [23]. After the first sample collection of a quantity just over 1 mL, a second sample was obtained through a syringe containing 0.5 mL of anticoagulant solution that was connected to the needle inserted in the coelomic cavity. The anticoagulant solution CCM 2× (NaCl 1 M, MgCl_2_ 10 mM, EGTA 2 mM, Hepes 40 mM, pH 7.2) had been added into the syringe ensuring a 1:1 ratio with the CF withdrawn [24]. Cell-free coelomic fluid (CF) was obtained from the supernatant of the coelomic fluid collected without an anticoagulant after centrifugation at 3000× *g* for 10 min at 4 °C while the coelomocytes were isolated by coelomic fluid with an anticoagulant after centrifugation at 600× *g* for 10 min at 4 °C.

### 2.5. pH and Osmolarity Evaluation in the Coelomic Fluid (CF)

The CF pH (pH_CF_) was measured with a Jenway 3020 pH meter (Jenway, London, UK). The CF osmolarity was measured with an osmometer (Digital Osmometer, Roebling, Berlin, Germany).

### 2.6. Coelomocyte Count and Morphological Typing

Total coelomocytes counts were made using a Neubauer-improved chamber under a light microscope (Apotome.2, Zeiss, Oberkochen, Germany). The values were corrected taking into account the dilution with anticoagulant. The count values were the average number of coelomocytes observed in five microscopic fields. The coelomocyte types were morphologically discriminated into phagocytes, red and white amoebocytes, and vibratile cells (Appendix A) [25,26]. A standard number of 2 × 10^5^ coelomocytes of each animal was seeded in a 96-well plate and incubated for 30 min before adding the fluorescent probes used to evaluate mitochondrial membrane potential (MMP), lipoperoxidation (LPO), and ROS production by these cells in the three groups compared.

### 2.7. Evaluation of Mitochondrial Membrane Potential (MMP)

JC1 (5,5′,6,6′-Tetrachloro-1,1′,3,3′-tetraethyl-imidacarbocyanine iodide) (Life Technologies, Milan, Italy) is a vital dye used to assess mitochondrial membrane potential. This molecule undergoes a reversible change in fluorescence emission shifting from ~595 nm wavelength (red peak), labeled as high mitochondrial membrane potential due to the formation of J-aggregates, to ~535 nm wavelength (green peak), labeled as low mitochondrial membrane potential due to the monomeric form of JC1 [10]. Aliquots of JC1 were prepared from a 7.7 mM stock solution in DMSO and stored at −20 °C. Before use, each aliquot was further diluted 1:5 with DMSO, and 1 µL of this solution was added to the 200 µL cell sample in natural filtered seawater (NFSW) within the 96-well plate. After 30 min incubation at 18 °C, the cells were washed by removing each well of the supernatant with a multi-channel pipette and adding fresh NFSW. CCCP (carbonyl cyanide 3-chlorophenylhydrazone) (Sigma-Aldrich, Milan, Italy) is a protonophore uncoupling oxidative phosphorylation in mitochondria and causes a decrease in mitochondrial membrane potential. Treatment of 2 µM CCCP was applied to the coelomocytes for the positive control. MMP was read with a microtiter plate reader (Infinite M1000 PRO; Tecan, Switzerland) and measured as the ratio (F0B/F0A) between the absolute fluorescence intensity produced by the red (F0B) and the green (F0A) wavelength emission peaks.

### 2.8. Evaluation of Lipid Peroxidation (LPO)

C11-BODIPY^581/591^ undecanoic acid (4,4-difluoro-5-(4-phenyl-1,3-butadienyl)-4-bora-3a,4a-diaza-s-indacene-3-undecanoic acid) (Life Technologies, Milan, Italy) is a fluorescent lipid peroxidation (LPO) sensor whose oxidation of the polyunsaturated butadienyl portion shifts the fluorescence emission peak from ~595 nm to ~520 nm wavelength [10]. A 1 mM C11 stock solution was prepared in DMSO and stored at −20 °C. Approximately 1 µL of this solution was added to the 200 µL cell sample in AFSW within the 96-well plate. After 30 min incubation at 18 °C, the supernatant of each well was aspirated by a multi-channel pipette, and fresh NFSW was added. At the time of C11 loading, the control coelomocytes were treated with 0.2 mM ferrous sulfate and 1 mM ascorbic acid and used as the positive control. The 96-well plate was read with a microtiter plate reader (see above) and the LPO was evaluated by relating the fluorescence peak values at ~520 nm wavelength (F0A) to the sum of the fluorescence peak values at the ~520 (F0A) and ~595 (F0B) nm emission wavelengths (F0A/(F0A + F0B) × 100).

### 2.9. Evaluation of Intracellular Hydrogen Peroxide Content (H_2_O_2_)

The intracellular content of hydrogen peroxide (H_2_O_2_) was assessed by using the 2′,7′-dichlorodihydrofluorescein diacetate (H_2_DCFDA) (Life technologies, Milan, Italy) [27]. This is a membrane-permeable, non-fluorescent, dye that, inside cells, is hydrolyzed by intracellular esterase into DCFH, which is oxidized by the intracellular H_2_O_2_ to the fluorescent compound DCF, whose fluorescence is proportional to the intracellular H_2_O_2_ content. A 1 mM H_2_DCFDA stock solution was prepared in DMSO and stored at −20 °C. Approximately 1 µL of this solution was added to the 200 µL cell sample in AFSW within the 96-well plate. After 30 min incubation at 18 °C, the supernatant in each well was removed by a multi-channel pipette and fresh AFSW was added. The samples were read with a microtiter plate reader (see above). The intracellular H_2_O_2_ levels were evaluated in arbitrary units (a.u.) on fluorescence emission peak intensity at ~525 nm wavelengths, setting the excitation wavelength at 488 nm. The positive controls were prepared by incubating the control coelomocytes with 25 μM hydrogen peroxide.

### 2.10. Respiratory Burst

At 3 days from the experiment’s start, the coelomocytes of the three groups were activated by the unspecific non-receptor protein kinase C (PKC) activator phorbol 12-myristate 13-acetate (PMA) (Sigma P1585). PMA stock solution was prepared in DMSO (1 mg mL^−1^) and stored at -80 °C. Respiratory burst was measured by fluorescence spectroscopy with a microtiter plate reader (see above) in either JC-1- or H_2_DCFDA-loaded cells. A standard number of coelomocytes was seeded in a 96-well plate (2 × 10^5^/well), loaded with either JC-1 or H_2_DCFDA (see Section 2.7 and 2.9) and treated with 10 µg PMA mL^−1^ [28]. The 96-well plate was read 1 min before and 1, 15, and 30 min after the PMA treatment. After reading at 1 and 15 min, the samples were incubated at 18 °C. 

### 2.11. Statistical Analysis

Statistical analyses were performed by ANOVA by using Systat 11.0 (Systat Software Inc., San Jose, CA, USA). Before the analyses, the percentage values were transformed in arcsine, whereas for pH_i_, the H+ concentrations were log transformed. Normal data distribution and homogeneity of variance were assessed by the Shapiro–Wilks test and Levene’s test, respectively. Pair-wise comparisons of the means were performed with Fisher’s least significant differences (LSD) test. The threshold of *p* < 0.05 was used as the minimum level of statistical significance. Data are presented as the mean ± standard deviation (SD).

## 3. Results

Small variations (16.56 ± 0.22, 22.71 ± 1.24, and 27.66 ± 0.91 °C) in the set and ambient (control) temperatures were recorded during the experiment by taking measurements twice a day. However, to facilitate the description, the results will referred to the set temperatures. 

### 3.1. Coelomic Fluid Parameters 

The pH_CF_ did not differ among the animals exposed to different temperature conditions as reported in Table 1, indicating that their acid–base status did not change after either 3 or 7 days of HS and remained different from the pH of the external environment (pH = 8.22 ± 0.06). Similarly, the CF osmolarity did not significantly differ among the experimental groups (Table 1) and remained similar to that of the external environment (1140 ± 3 mOsm) (Table 1). 

### 3.2. Immune Cell Number and Cell Type Distribution

The total number of coelomocytes did not significantly differ between the experimental groups after either 3 or 7 days of HS (Table 1). As shown in Figure 1, phagocytes were the most abundant immune cells in all groups. However, in the coelomocytes recovered from sea urchins exposed for 7 days to 28 °C, the percentage of phagocytes (81.0 ± 4.7%) significantly (*p* < 0.05) decreased compared to those harvested from the control (90.2 ± 2.1%) and 23 °C-exposed (90.4 ± 2.1%) sea urchins (Figure 1B). The red amoebocytes showed a significant (*p* < 0.05) temperature-depended trend increasing from 1.1 ± 0.2% (17 °C) to 2.7 ± 0.2% (23 °C) and 2.2 ± 0.7% (28 °C) after 3 days (Figure 1A) and from 1.2 ± 0.4% (17 °C) to 3.9 ± 1.0% (23 °C) and 14.3 ± 3.6% (28 °C) after 7 days (Figure 1B) from the start of the experiment. Conversely, the white amoebocytes showed a significant (*p* < 0.05) inverse temperature-depended trend decreasing from 7.2 ± 1.1% (17 °C) to 5.0 ± 0.6% (23 °C) and 4.1 ± 1.4% (28 °C) after 3 days (Figure 1A) and from 6.6 ± 1.3% (17 °C) to 3.9 ± 0.8% (23 °C) and 2.5 ± 1.0% (28 °C) after 7 days (Figure 1B) from the start of the experiment. In addition, the group exposed to 28 °C further significantly (*p* < 0.05) differentiated both the red and white amoebocyte content from the group exposed to 23 °C only after 7 days of treatment (Figure 1B). The dynamic change in the incidence of these two cell types on the total coelomocytes population in response to HS resulted in a progressive increase in the red/white amoebocyte ratio along with the temperature rise from 17 °C (0.15 and 0.18%) to 23 °C (0.5 and 1.0%) and 28 °C (0.5 and 5.8%) after 3 and 7 days from the start of the experiment, respectively. 

### 3.3. Mitochondrial Membrane Potential (MMP) in the Coelomocytes

The mitochondrial activity of the coelomocytes, evaluated as mitochondrial membrane potential (MMP), was significantly affected by HS. In fact, a significant (*p* < 0.05) increase in the MMP value was recorded in the group at 23 °C at 3 days following HS induction (24.0 ± 9.7 vs. 14.7 ± 5.6) while a significant (*p* < 0.05) decrease was detected in the group exposed to 28 °C for both 3 (6.4 ± 2.9 vs. 14.7 ± 5.6) and 7 (11.3 ± 3.2 vs 16.6 ± 7.9) days (Figure 2A). A significant (*p* < 0.05) lowering of the MMP value was also observed in the positive control group treated with CCCP.

### 3.4. Lipid Peroxidation (LPO) in the Coelomocytes

The LPO in the coelomocytes, evaluated by fluorescent dye C11-Bodipy, significantly (*p* < 0.05) increased only after 7 days of exposure to 28 °C (64.7 ± 5.8 vs. 54.7 ± 7.9) (Figure 2B). A significant (*p* < 0.05) increase in the LPO values was also observed in the positive control treated with FeSO_4_ and ascorbic acid.

### 3.5. Hydrogen Peroxide (H_2_O_2_) Content in the Coelomocytes

A significant (*p* < 0.05) decrease in H_2_O_2_ content was found in the coelomocytes at both 23 °C (192 ± 57 and 285 ± 21 a.u.) and 28 °C (172 ± 57 and 269 ± 29 a.u.) vs. 17 °C (255 ± 40 and 332 ± 26 a.u.) after 3 and 7 days from the start of the experiment, respectively (Figure 2C). Treatment with H_2_O_2_ produced a significant increase in fluorescence intensity in the positive controls. 

### 3.6. Respiratory Burst in the Coelomocytes

Coelomocyte stimulation with PMA was evaluated by monitoring changes in MMP and H_2_O_2_ content in the 30 min following treatment (Figure 3). The MMP shows a significant and progressive decrease in all the experimental groups, reducing the differences between them as the distance from the treatment increases. However, significant differences between the group kept at 23 and 28 °C were found throughout all the evaluation periods. Conversely, the H_2_O_2_ content shows a progressive increase in all three experimental groups with a progressive and significant divergence between the group at 17 °C and the two groups under HS as the time from the treatment increases.

## 4. Discussion

The rapid and progressive increase in sea temperatures related to climate change is a serious challenge to the health of marine ecosystems [29]. Temperature is crucial for the biological processes of all organisms and its increase alters many vital functions and threatens the survival of many species [30]. The sea urchin *P. lividus* represents an efficient bioindicator of environmental contamination [31]; it might be expected to be gradually damaged by extreme anthropogenic pressures especially related to temperature rise but also to ocean acidification and pollution [32,33,34]. With that in mind, we aimed to deeply understand how sea urchins respond to the increase in seawater temperature with a special focus on immune cells. The data obtained from this study demonstrated in the sea urchin *P. lividus* that short-term HS significantly affects both the distribution of the coelomocyte type populations and several physiological parameters of these cells such as mitochondrial activity, lipid peroxidation, H_2_O_2_ content as well as the respiratory burst. However, short-term HS did not alter the CF osmolarity and pH. The CF osmolarity was maintained at the same values of those recorded in the seawater, confirming the poor ability of many species of echinoderms to regulate ion concentration in their extracellular fluids [35]. Instead, the pH_CF_ was maintained at a lower level than seawater and showed a low variability either within or between the experimental groups. A lower pH_CF_ than that of seawater has been attributed to a result of CO_2_ retention (slow diffusion rate) and due to the accumulation of acidic metabolites [36]. The effects of temperature on pH_CF_ are controversial. In *P. lividus*, a significant relationship between pH_CF_ and seawater pH was reported but not associated with temperature variations [37]. Conversely, in the sea urchin *Arbacia punculata* [11], either moderate (28 °C) or intense (32 °C) short-term HS caused a significant decrease in pH_CF_. Our results confirm those already found in *P. lividus*, and demonstrate differences between *P. lividus* and *A. punctulate*; the former, in fact, together with lower thermotolerance, shows upper pH_CF_ values of about 0.6–0.7.

Sea urchin coelomocytes engage in various activities ranging from immune to metabolic functions with the transport of nutrients and ensuring gaseous exchanges [15]. These activities are distributed among the different cell types making up the coelomocyte population. In the present study, large variability was found in the coelomocyte concentration, which was not significantly affected by short-term HS. This number undergoes a significant increase in the presence of immunological challenges [38]. No significant change in total concentration was found under chemical stress such as that induced by marine pollution [39]. However, a study on the effect of future seawater pH and temperature variations in the climate change scenario showed a significant lowering of the total concentration of coelomocytes in the sea urchins after 30 days but this was not detected at 15 days of exposure to 23 °C [40]. The cellular morphological profile confirms previous studies [24] and attests to phagocytes as a primary component of the coelomocyte population. The significant reduction in this cell type detected in animals exposed to 28 °C for 7 days could probably be traced back to the significant increase in the number of red amoebocytes, normally present in a minority ratio compared to phagocytes. The increase in red amoebocytes together with the increase in temperature is in agreement with previous studies [19] and occurs simultaneously with the reduction in white amoebocytes. Worthy of consideration is the ratio between these two cell types, which, in the present study, grows exponentially with increasing stress intensity. These differences are evident as early as 3 days after either moderate (23 °C) or intense (28 °C) HS. This finding supports the use of this ratio as a biomarker for the welfare evaluation of sea urchins [14,41].

Phagocytosis, encapsulation, and the production of ROS and RNS are the most widespread mechanisms by which the immune cells of invertebrates react under conditions of defense against pathogens and stressors [42]. In the sea anemones *Exaiptasia diaphana* and *Nematostella vectensis*, overnight exposure to 30 °C (moderate HS) increased phagocytic activity and immune gene expression together with cellular ROS production compared to the ambient temperature (18 °C) [43]. In the present study, we focused on the production of ROS and on the effects related to it, such as lipid peroxidation. Besides representing an important marker of cellular welfare, mitochondrial activity is also closely associated with the production of ROS [44,45]. MMP, as a marker of mitochondrial activity, has been widely used for evaluating the quality of gametes [46] but also the atresia grade in ovarian follicles [47] as well as numerous diseases, including diabetes, cancer, neurodegenerative diseases, and ischemia–reperfusion injury [48]. The results of the present study confirm what has already been found on the effects of long-term (30-d) HS in mussel *M. galloprovincialis* spermatozoa [10], in which MMP increased during the first two weeks of high thermal exposure and then collapsed. The increase in MMP in the first period of HS is an adaptive response. It represents the effort that an organism makes to compensate for a stressful situation, i.e., its resilience. If resilience fails to move the homeostasis of the organism towards the new level imposed by the stressful situation, degenerative mechanisms take over. Our previous study described these consequences and attested them by morphological alterations affecting the mitochondria [10]. In the present study, it is clear that already after three days of exposure to severe (28 °C) HS conditions, the cells fail to reach the resilience status. The lowering of mitochondrial activity marks this occurrence although a rush of mitochondrial activity might also occur earlier than three days in the coelomocytes of animals exposed to 28 °C and before collapsing. In support of this hypothesis, there are studies of very short HS in green-lipped mussel *Perna canaliculus* in which exposure at 28.5 °C for 20 min, far above the thermal comfort zone at 7.3 °C, produced a significant increase in ROS production in hemocytes [49]. However, this production drastically collapsed when the animals were exposed to 30 °C for 60 min.

LPO is a marker of chronic oxidative stress [50]. It is closely associated with the ROS increase, which, by overcoming the cell antioxidant mechanisms, causes membrane damage and several dysfunctions. This parameter has been also widely used to evaluate the quality of gametes [46] as well as pathologies such as diabetes mellitus, atherosclerosis, and chronic inflammation [50]. It can be estimated using various methods, the most common of which is malondialdehyde analysis by the thiobarbituric acid reactive substances (TBARS) assay [51]. In the gill tissue of the green mussel *Perna viridis*, malondialdehyde content and MMP increased significantly under both cold and heat stress [52]. In our study, using a fluorescence spectrometry method, we recorded a late increase in this parameter 7 days after the induction of intense HS. This occurrence was delayed in comparison to mussel spermatozoa [10] in which LPO was significantly higher than in the control group already two days after HS induction. We do not have a clear explanation for this finding; most likely, the higher content of antioxidant substances in coelomocytes compared to spermatozoa could be the basis of this result.

The increase in the H_2_O_2_ content takes on a different meaning in immune cells from that detectable in other somatic cells for which it represents a marker of acute oxidative stress [53]. Coelomocytes, as well as granulocytes in mammals, use the production of H_2_O_2_ as one of the defense mechanisms against pathogens [16]. Therefore, the significant reduction in the H_2_O_2_ content in the coelomocytes as the HS intensity increases both at 3 and at 7 days of HS could be associated with a lowering of the immune defenses of these cells. This hypothesis is further supported by the results obtained from the respiratory burst test. Respiratory burst, induced in the present study by PMA, is a strategy by which immune cells produce large quantities of H_2_O_2_ to contrast pathogens. Respiratory burst does not appear to be associated with mitochondrial activity, as found in human leukocytes [54], although prolonged (2-h) incubation with oligomycin, a mitochondrial ATPase inhibitor, resulted in an impaired ability of neutrophils to activate respiratory burst and also inhibited chemotaxis [55]. In the present study, this finding was confirmed in the sea urchin coelomocytes. In fact, it is clear that following respiratory burst induction, in the face of the decrease in mitochondrial activity, there was a progressive increase in the production of H_2_O_2_. Although the coelomocytes of sea urchins under HS still responded to PMA treatment, the lower hydrogen peroxide produced compared to that obtained in the coelomocytes of animals kept under thermo-comfort conditions suggests lower immune cell function associated with heat stress.

## 5. Conclusions

This study provides a new insight into the effects of the increasing temperatures on a species already heavily compromised by global warming, the sea urchin *P. lividus*, which holds the structure and the control of macroalgae assemblages and thus the shape of the benthic Mediterranean seascape [56]. In detail, short-term heat stress significantly affected the immune cells of the female *P. lividus* sea urchins. Immune cell dysfunctions were significantly detectable already after 3 days with certain biomarkers, such as cell type distribution, mitochondrial activity, and hydrogen peroxide production, even though they became evident at 7 days of heat stress. At this time, all the above biomarkers together with coelomocyte lipid peroxidation showed marked differences between the heat stressed and control animals. These findings, together with the lower response to the experimentally induced respiratory burst, suggest that, in the females of sea urchins, the immune response is early compromised in the course of heat stress and aggravates the implications of climate change on the fitness of marine species.

## Figures and Tables

**Figure 1 animals-13-01954-f001:**
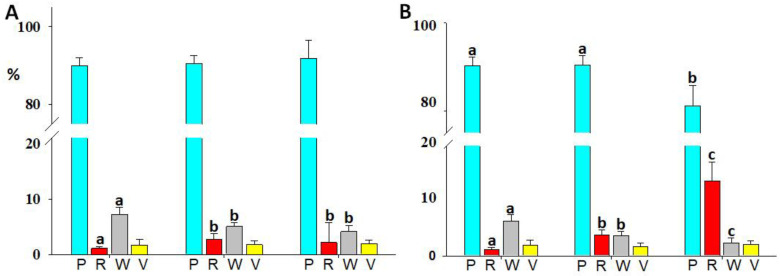
Immune cell type distribution (%, mean ± SD) in sea urchins after 3 (**A**) and 7 (**B**) days of exposure at 17, 23, and 28 °C. (P) Phagocytes; (R) red amoebocytes; (W) white amoebocytes; (V) and vibratile cells. Different letters (a, b, and c) mark statistically significant (*p* < 0.05) differences within the cell types and between the experimental groups.

**Figure 2 animals-13-01954-f002:**
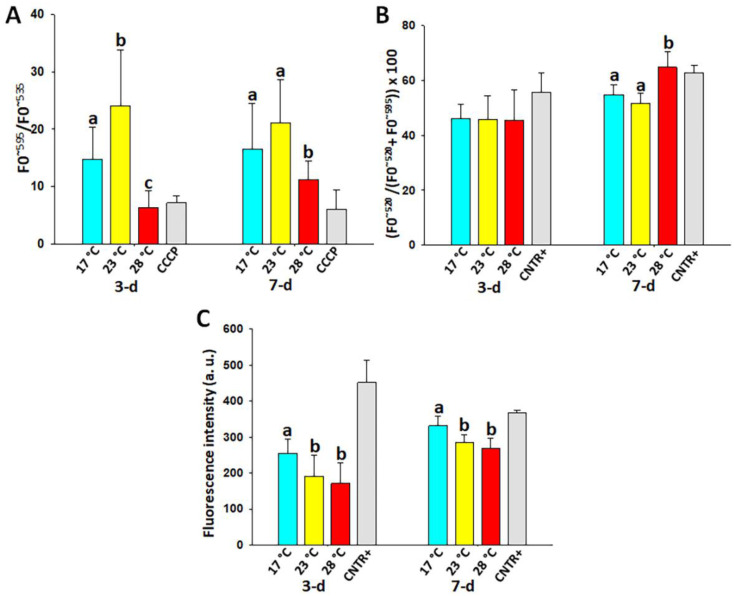
Mean (±SD) values of mitochondrial membrane potential (**A**), lipid peroxidation (**B**), and hydrogen peroxide content (**C**) in the coelomocytes of the sea urchin specimens exposed to 23 and 28 °C and to a thermal comfort temperature (17 °C) for 3 and 7 days. Statistically significant (*p* < 0.05) differences within each time of exposure are marked with a, b, and c.

**Figure 3 animals-13-01954-f003:**
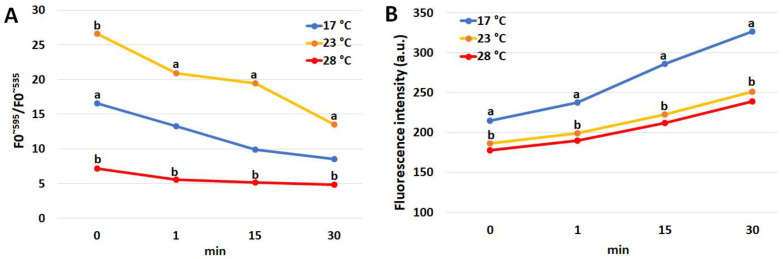
Mean values of mitochondrial membrane potential (**A**) and hydrogen peroxide content (**B**) in the coelomocytes of sea urchin specimens (*n* = 6) exposed to 23 and 28 °C and to a thermal comfort temperature (17 °C) for 3 days and treated with 10 µg PMA mL^−1^. Time 0 = 1 min before PMA treatment; time 1, 15 and 30 = 1, 15, and 30 min after PMA treatment, respectively. Statistically significant (*p* < 0.05) differences with values at 28 °C are marked with a and b.

**Table 1 animals-13-01954-t001:** Celomatic fluid (CF) pH (pH_CF_) and osmolarity and coelomocyte concentration in the sea urchins exposed for either 3 or 7 days to 17, 23, and 28 °C.

	Time of Exposure (Days)	Animals *n*	Coelomocyte Concentration (×10^6^/mL)	pH_CF_	CF Osmolarity (mOsm)
17 °C	3 7	6 6	10.0 ± 5.6 6.9 ± 3.3	7.87 ± 0.03 7.83 ± 0.09	1143 ± 2 1144 ± 3
23 °C	3 7	6 6	11.6 ± 2.6 7.5 ± 2.9	7.84 ± 0.09 7.86 ± 0.10	1149 ± 7 1139 ± 9
28 °C	3 7	6 6	8.8 ± 5.5 7.6 ± 3.2	7.82 ± 0.07 7.82 ± 0.18	1144 ± 2 1138 ± 10

## Data Availability

Not applicable.

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
