# Peer review of "Short-Term Thermal Stress Affects Immune Cell Features in the Sea Urchin Paracentrotus lividus"

_animals, 2023, doi:10.3390/ani13121954_

Round 1

Reviewer 1 Report

The manuscript entitled "Short-term thermal stress affects immune cell features in the sea urchin Paracentrotus lividus" deals with an interesting and important topic although it is already much studied in the literature on various species.

There are several critical points that I hope the authors will be able to improve or modify. The most complicated thing is the experimental plan as it was conceived and implemented. Below I detail my doubts about this manuscript.

The abstract is well written but I recommend changing the final part of the results. It would be important to highlight and summarize the results obtained and not detail the changes at various temperatures or days, this must be done in the results. Furthermore, the conclusions should be moderate, to say that the temperature has these effects... the experimental plan had to include other variables (e.g. variations in the availability of food, pH, salinity etc)!!

The keywords should not contain words already inserted in the title.

As for the introduction, it is fine for the authors to highlight their studies to justify the reasons of this study…but the introduction is a summarized state of the art and for this reason also other study in the literature testing the effects of different temperatures on cellular and non-cellular responses of marine organisms should be mentioned and treated. This would give value to the manuscript and to the authors' knowledge of the subject itself. The introduction is too poor in my opinion and does not justify the study undertaken.

Materials and methods. The authors should detail the feeding methods of the individuals and above all whether or not they were fed during the experimental plan and before sampling (the relationship between feeding and the parameters evaluated on the coelomic fluid is known in the literature).

The sex of sea urchins can be known normally by inducing the emission of gametes so the biggest problem in my opinion is to be sure that they were all female. I'm not sure that the method reported in the literature (reading the study) gives certainty or reduces stress.... the animals kept out of the water to be observed under the microscope would be induced to emit gametes. Furthermore, the motivation that the authors provide for choosing to use only females is not satisfactory. If the reason was to avoid reproductive interactions, the males could be placed in separate tanks. Also why avoid interactions of a reproductive nature and therefore create unrealistic conditions?

In my view it is not possible to indicate stable temperatures at which the animals were kept without standard deviation.

“the animals were not fed in order not to pollute the tanks with manure that would have involved cleaning operations compromising the rearing environment and causing a further stress source” is not an adequate justification, an adequate diet would not have dirty the tank. Furthermore, also in this case an important variable is excluded which does not allow to reach certain conclusions already described in the abstract

“After 3 days, three animals from each experimental group were sampled and, then, moved to other tanks at 17 °C” I got lost..... really the experimental plan as described is not clear.

The authors collect coelomic fluid using the needle syringe and it is known that this can influence the result and the immune responses of the sea urchin. This type of coelomic fluid sampling can influence the results obtained.

It makes no sense to test two different samples with and without anticoagulant. The authors could choose to work either with or without anticoagulant throughout with the necessary corrections to the protocol known in the literature.

“P” in statistics should be lowercase

In my opinion, it would be appropriate to include photos of the observed cell types. Furthermore, slides with specific stains would have helped in a more accurate differential count.

The graphs are not homogeneous in formatting as well as in my opinion being structured in an unprofessional manner. They are not acceptable for a scientific publication.

In the discussions the authors state "Nowadays, it needs to underline how temperature is crucial for the biological processes of all organisms and how its increase alters many vital functions and threatens the survival of many species" I think it is important to give valid reasons to this sentence.

Why is the role of sea urchins as bioindicators not highlighted in any part of the manuscript?

The discussions in general are well written even if they are clearly lacking (like the introduction) of other scientific literature concerning the study of the effects of temperature changes on other marine invertebrates.

The conclusions should be moderate because, as I said previously, certain results cannot be given if other variables involved, different sexes are not considered and realistic mesocosms are not reproduced

I suggest moderate editing of English language

Author Response

We thank the reviewer very much for the work done. Below are the point-by-point responses to the criticisms raised.

The manuscript entitled "Short-term thermal stress affects immune cell features in the sea urchin Paracentrotus lividus" deals with an interesting and important topic although it is already much studied in the literature on various species.

There are several critical points that I hope the authors will be able to improve or modify. The most complicated thing is the experimental plan as it was conceived and implemented. Below I detail my doubts about this manuscript.

Reply. We thank the reviewer for the analysis conducted on our manuscript and we hope, thanks to it, to be able to improve the quality of our paper and clarify the doubts raised.

The abstract is well written but I recommend changing the final part of the results. It would be important to highlight and summarize the results obtained and not detail the changes at various temperatures or days, this must be done in the results. Furthermore, the conclusions should be moderate, to say that the temperature has these effects... the experimental plan had to include other variables (e.g. variations in the availability of food, pH, salinity etc)!!

Reply. We accepted the suggestion and modified the conclusions. As regards the numerous variables that could affect the results, we consider this aspect in the Introduction (L50-60) and, in our experimental design, we have tried to minimize sources of variation, such as (i) salinity, by supplying seawater collected directly from the sea, conveyed to collection tanks and filtered; (ii) pH, by using in the three groups the same seawater that was daily monitored; (iii) the availability of food, by not supplying food during the experimentation days, relying on the enormous experience gained in the animal facility of the SZN in Naples, where the experiment took place. Further attention to reducing the sources of variability was to use all individuals of the same sex gender, considering that, in previous studies, molecular analyses have highlighted a gender-specific response to stress conditions.

The keywords should not contain words already inserted in the title.

Reply. Thanks for the note. We have changed the keywords accordingly.

As for the introduction, it is fine for the authors to highlight their studies to justify the reasons of this study…but the introduction is a summarized state of the art and for this reason also other study in the literature testing the effects of different temperatures on cellular and non-cellular responses of marine organisms should be mentioned and treated. This would give value to the manuscript and to the authors' knowledge of the subject itself. The introduction is too poor in my opinion and does not justify the study undertaken.

Reply. We have expanded the introduction, as required, reducing the details of the studies cited and providing a detailed overview of the studies conducted on this topic.

Materials and methods. The authors should detail the feeding methods of the individuals and above all whether or not they were fed during the experimental plan and before sampling (the relationship between feeding and the parameters evaluated on the coelomic fluid is known in the literature).

The sex of sea urchins can be known normally by inducing the emission of gametes so the biggest problem in my opinion is to be sure that they were all female. I'm not sure that the method reported in the literature (reading the study) gives certainty or reduces stress.... the animals kept out of the water to be observed under the microscope would be induced to emit gametes. Furthermore, the motivation that the authors provide for choosing to use only females is not satisfactory. If the reason was to avoid reproductive interactions, the males could be placed in separate tanks. Also why avoid interactions of a reproductive nature and therefore create unrealistic conditions?

Reply. As previously reported and requested by the reviewer, we deemed it necessary to reduce the sources of variation and nutrition is certainly one of them. In accordance with the opinion of the technicians of the SZN animal facility, we have decided not to provide food to the animals, a condition that these animals are perfectly able to face. This could cause a moderately stressful situation which would, however, be common to all three experimental groups, and, hence, normalized in the experimental design.

Regarding the sexing of animals, a solid method of sexing has been developed in our Institute (Brundu et al., Aquaculture 2023) which works very well and is based on previous studies by Tahara et al. (1958 and 1960) that discriminated the genital papillae in several echinoderm species as conical protuberances in males, while stumpy and often flat and sunk below the body surface in females. This technique does not induce gamete spawning in a higher percentage than all other operations of collection, movement, and manipulation of the animals. We have amply justified this choice and indicated this aspect in the title of the manuscript. Based on the experimental model, having a limited number of animals, we believe that it was more profitable to provide clear indications rather than variable and not solid results. We also recall that in the literature, studies conducted on a single sexual gender abound in other species. Our own group has published a study on thermal stress in Mytilus galloprovincialis considering only the effects on male gametes (Boni et al., MRD 2016) but many examples can be found in the literature related to reproduction, such as in oysters (https://doi.org/10.1002/mrd.23268), octopus (https://doi.org/10.3389/fphys.2018.01920), Pacific abalone ( https://doi.org/10.3389/fmars.2021.664426) as well as in immune cells, such as  https://doi.org/10.1152/ajpregu.1999.276.1.R97; https://doi.org/10.1210/en.2014-1794; https://doi.org/10.3389/fimmu.2021.659469.

In my view it is not possible to indicate stable temperatures at which the animals were kept without standard deviation.

Reply. In the Results (L240-3), we added this information.

“the animals were not fed in order not to pollute the tanks with manure that would have involved cleaning operations compromising the rearing environment and causing a further stress source” is not an adequate justification, an adequate diet would not have dirty the tank. Furthermore, also in this case an important variable is excluded which does not allow to reach certain conclusions already described in the abstract

Reply. We already replied to this criticism (see above)

“After 3 days, three animals from each experimental group were sampled and, then, moved to other tanks at 17 °C” I got lost..... really the experimental plan as described is not clear.

Reply. We modified the text to make clearer the experimental design.

The authors collect coelomic fluid using the needle syringe and it is known that this can influence the result and the immune responses of the sea urchin. This type of coelomic fluid sampling can influence the results obtained.

Reply. The collection of coelomic fluid (CF) was carried out at the end of the experimental period foreseen for the considered animals. In practice, after having taken the CF, the animals had finished their experimental use and were moved to other tanks of the enclosure for breeding. The collection of CF by syringe is described in many papers for immunology studies (https://doi.org/10.1016/0022-2011(78)90170-2; https://doi.org/10.1016/0145-305X(91)90018-T; https://doi.org/10.1016/j.cbpa.2012.11.021) and we have not found any criticality related to this method. Actually, we have found no alternative to this for CF sampling.

It makes no sense to test two different samples with and without anticoagulant. The authors could choose to work either with or without anticoagulant throughout with the necessary corrections to the protocol known in the literature.

Reply. The withdrawal of coelomic fluid without using an anticoagulant was mandatory to evaluate the pH and osmolarity. In contrast, the anticoagulant liquid was necessary for the count and management of coelomocytes, avoiding their coagulation. This is widely described in the literature, such as https://doi.org/10.1016/0014-4827(78)90185-4; https://doi.org/10.1016/j.dci.2014.11.013; https://doi.org/10.1016/bs.mcb.2018.11.009.

“P” in statistics should be lowercase

Reply. Done

In my opinion, it would be appropriate to include photos of the observed cell types. Furthermore, slides with specific stains would have helped in a more accurate differential count.

Reply. We added this information in the Supplementary materials (Figure S1) and cited in the text on L170.

The graphs are not homogeneous in formatting as well as in my opinion being structured in an unprofessional manner. They are not acceptable for a scientific publication.

Reply. we redid the graphs using the same style.

In the discussions the authors state "Nowadays, it needs to underline how temperature is crucial for the biological processes of all organisms and how its increase alters many vital functions and threatens the survival of many species" I think it is important to give valid reasons to this sentence.

Reply. We added the reference (#30, L315).

Why is the role of sea urchins as bioindicators not highlighted in any part of the manuscript?

Reply. We have added this information (# 31, L316).

The discussions in general are well written even if they are clearly lacking (like the introduction) of other scientific literature concerning the study of the effects of temperature changes on other marine invertebrates.

Reply. We have expanded the discussion by inserting new citations.

The conclusions should be moderate because, as I said previously, certain results cannot be given if other variables involved, different sexes are not considered and realistic mesocosms are not reproduced

Reply. We have changed the conclusions in the Abstract, at the end of the Discussion, and in the Conclusions. We believe that we have been very cautious and that we have complied with the results obtained.

Reviewer 2 Report

The concept of the study is quite interesting.

My comments focus mostly around the use of English, which in my opinion needs a little bit more work. At some points the text is written with very much detailed description and ends up being very complex and difficult to read and to follow. 

For more details, see the comments on the attached pdf.

My suggestion is for a major revision of the text, but since it is not available as an option I am selecting the "Accept after minor revision" option

Already commented on the Quality of English

Author Response

We thank the reviewer very much for the work done. We have accepted all suggestions and modified the text accordingly.

Round 2

Reviewer 1 Report

The manuscript has been greatly improved even if in my opinion the literature cited is still very scarce.

Author Response

We have made further modifications and additions to our paper hoping to have gone in the right direction and to have improved the opinion of the reviewer who we thank for the valuable suggestions.

Reviewer 2 Report

The authors complied with the majority of the comments/suggestions from the reviewers.

Still, in some cases, the result was not an actual improvement. For example, in lines 88 - 98, in the Introduction, the authors are again over-discussing some of the results from previous studies in an attempt to present the state-of-the-art, but they just make the comprehension of the whole paragraph more complicated.

The experimental design (lines 134 -147) was not very clear and they should have tried to write it again from scratch, in a more simple manner. Instead they just rephrased some parts, which in my opinion, did not help so much with un-complicating the text.

In line 141, (I forgot to mention this in the previous round of the review) using  the word "manure" is not very accurate. "Manure" is more commonly used to decribed the animal waste used as a fertilizer in agriculture. Maybe "faeces" or "feces" would be more appropriate.

The use of the word "Panel" in figures, in order to distinguish among the different graphs is also ccomplicating the reading and comprehending the Captions. Just use the common way of citing different graphs by using parentheses i.e. (A), (B), (C)

The Results are still presented in a complicated manner. 

In line 266, differences do not "emerge", they are "detected" 

The Discussion was enriched with some minor additions, but still the authors could elaborate more, even by moving here and discussing the results from the previous studies, which they mentioned in the Introduction and their potential correletion or differentiation from their current findings.

All in all, the manuscript was improved in a general sense, but still I can't say I am 100% satisfied by the revised version.

I

Author Response

The authors complied with the majority of the comments/suggestions from the reviewers.

Reply. We thank the reviewer for taking the time to review our paper and for helping us improve its form and content.

Still, in some cases, the result was not an actual improvement. For example, in lines 88 - 98, in the Introduction, the authors are again over-discussing some of the results from previous studies in an attempt to present the state-of-the-art, but they just make the comprehension of the whole paragraph more complicated.

Reply. We further removed not pertinent information making, we hope so, the text more readable.

The experimental design (lines 134 -147) was not very clear and they should have tried to write it again from scratch, in a more simple manner. Instead they just rephrased some parts, which in my opinion, did not help so much with un-complicating the text.

 Reply. We further modified this chapter.

In line 141, (I forgot to mention this in the previous round of the review) using  the word "manure" is not very accurate. "Manure" is more commonly used to decribed the animal waste used as a fertilizer in agriculture. Maybe "faeces" or "feces" would be more appropriate.

Reply. We replaced the word “manure” with “feces and food residues”

 The use of the word "Panel" in figures, in order to distinguish among the different graphs is also ccomplicating the reading and comprehending the Captions. Just use the common way of citing different graphs by using parentheses i.e. (A), (B), (C)

Reply. We modified the Figure legends as suggested.

The Results are still presented in a complicated manner. 

Reply. We rewrote the hard-to-follow part of the results. Unfortunately for legibility but fortunately for the scientific meaning, there are many significant results that must be reported and indicated numerically to describe the figure in detail.

In line 266, differences do not "emerge", they are "detected" 

Reply. Done.

The Discussion was enriched with some minor additions, but still, the authors could elaborate more, even by moving here and discussing the results from the previous studies, which they mentioned in the Introduction, and their potential correletion or differentiation from their current findings.

Reply. As suggested, we have reported in the discussion part of what was found in a previous study on M. galloprovincialis. However, since this study analyzed spermatozoa and shared only a few indicators with the present research, the possible comparisons with the results of the current study are few. However, they have been reported and compared with the results of the present study in many parts of the paper.